# Curvature as an Organizing Principle of Mid-level Visual Representation: A Semantic-preference Mapping Approach

**Shi Pui Donald Li**\*
Department of Cognitive Science
Johns Hopkins University
Baltimore, MD 21218
sli97@jhu.edu

**Michael F. Bonner**
Department of Cognitive Science
Johns Hopkins University
Baltimore, MD 21218
mfbonner@jhu.edu

## Abstract

A central challenge in visual neuroscience is understanding the mid-level representations of the ventral stream. We used a novel, data-driven approach (semantic-preference mapping) combined with an image-statistics approach (curvature index) to characterize the mid-level features of category-selective visual regions. First, we fit voxelwise encoding models using a deep convolutional neural network (DCNN) to predict scene-evoked fMRI responses in object-selective and scene-selective regions. We then performed semantic-preference mapping to examine how the responses of these encoding models changed when specific object classes were removed from natural images. This analysis motivated the hypothesis that object-selective cortex model is best predicted by mid-level features with curved contours, while scene-selective cortex model is best predicted by mid-level features with rectilinear contours. We further developed an image-computable model that outputs a summary statistic for the prevalence of curved contours in local image patches, and we used this model to demonstrate the importance of curvature-preferences for linking DCNN representations with the representations of category-selective cortex models. Overall, our findings suggest that curvature is a key property of the mid-level representations that are shared between DCNNs and category-selective cortex models of the ventral visual stream.

## 1 Introduction

Understanding the mid-level representation of the cortical visual hierarchy is essential for completing the puzzle of how sensory inputs are transformed to support goal-directed behaviors. In fact, recent studies have suggested that mid-level visual features may have a direct role in mediating the higher-level cognitive functions of visual cortex [15, 14, 3]. However, our understanding of mid-level representations remains far from complete. Mid-level features are particularly challenging to study because they reflect highly nonlinear transformations of image inputs and often do not correspond to intuitive concepts that can be easily described in words [20]. For these reasons, attempts at directly visualizing mid-level representations through feature-visualization methods applied to DCNNs or neural data often yield complex, uninterpretable patterns [24, 2, 23].

Here we sought to overcome the challenges of understanding mid-level representations by striking a balance between data-driven and hypothesis-driven experiments. We developed a method called semantic-preference mapping that systematically explores how DCNN representations are modulated by specific classes of objects or visual features. This approach is data-driven in that it allows one to

---

\*Corresponding author: https://pages.jh.edu/ sli97/

explore DCNN selectivity across a large and diverse set of natural images, but it is also constrained in that the selectivity profiles are defined over an experimenter-selected set of stimulus properties (e.g., semantic classes). We used semantic-preference mapping to explore the mid-level representations of category-selective visual cortex in the human brain. We first built DCNN encoding models to predict fMRI responses to natural images in a large-scale dataset [6]. We then examined the semantic preferences of DCNN encoding models for object- and scene-selective visual regions, which led us to hypothesize that curvature is a prominent organizing principle of the mid-level feature preferences in these regions. We further validated that curvature is a key property of these mid-level representations using a computational model of curvature tuning. Our findings suggest that curvature is an important property of mid-level representations that not only relates to the organization of visual cortex encoding models into category-selective regions but also appears in the tuning properties of DCNNs that are commonly used as computational models of human visual cortex.

## 2 DCNN encoding model

Previous studies have shown that supervised DCNNs are the best class of computational models for explaining representations in visual areas, and category-selective regions are often best explained by the mid-level layers of DCNNs, like AlexNet [10, 3]. DCNNs thus have the potential to help scientists explore the nature of mid-level visual representations in the human brain [7]. In this study, we utilized a voxelwise encoding model approach [17] and DCNNs to explore the mid-level representations of two category-selective regions in the ventral stream: the scene-selective parahippocampal place area (PPA) and the object-selective lateral occipital complex (LOC).

### 2.1 Model architecture

We employed a pretrained AlexNet trained on ImageNet [13] for our encoding models. First, an image was passed through AlexNet to obtain the feature maps of the target convolutional layer. Then global max-pooling was performed over the whole feature map for each channel to output a vector of feature activations. We included global max-pooling because we were specifically interested in examining feature selectivity rather than retinotopic biases. Object- and scene-selective regions have large receptive fields [22], so we predicted that our global-pooling procedure would have little impact on model performance while having the benefit of heavily reducing the number of model parameters. We confirmed this by comparing our results with those obtained without global pooling and found that the results were highly similar. After global pooling, the resulting vector of AlexNet feature activations was linearly connected to a set of output units that corresponded to predicted voxel activations. A separate encoding model was built for each voxel and each convolutional layer of AlexNet. See Fig.1A for illustration.

### 2.2 Model training

DCNN encoding models were trained on the BOLD5000 dataset [6]. This is a large-scale fMRI dataset in which each of 4 subjects viewed 5000 images from the ImageNet, COCO and Scene datasets. Only a very small subset of images ( 100-114 images) were repeated in the dataset, and noise ceilings were obtained by the mean correlation of the fMRI activation of the repeated images. During the training process, only the linear connection layer was trained using LASSO regression (L1 penalized regression). The motivation for using LASSO was to make the model more interpretable by assuming sparse coding of AlexNet features. However, we also observed that LASSO regression outperformed both ordinary least squares and ridge regression (see Appendix A). To assess the performance of the encoding models, we used a 10-fold cross-validation procedure over all images and computed the voxelwise correlations between the predicted and actual activations on each fold. We then took the mean of these correlations across all 10 folds to obtain a final prediction score for each voxel. Fig.1B illustrates the mean performance across subjects of the voxelwise encoding models for each ROI. Since not all images are presented repeatedly in the BOLD5000 dataset, noise ceilings are calculated based on the image subset that was presented more than once in the experiment. Therefore, the noise ceiling estimated could be biased to that image subset. The performance of the models exceeding noise ceiling by a significant margin suggests that the encoding models provide a meaningful fMRI activation predictions and reach the theoretical ceiling for model performance after accounting for the noise in the data. We found that convolutional layer 5, a middle layer of

AlexNet, performed the best in our two category-selective ROIs. Therefore, our follow-up analyses focus on layer 5. Before performing follow-up analyses, we re-trained layer 5 encoding models for all images using the entire BOLD5000 dataset. These encoding models served as a simulation of the corresponding visual cortex and we will call the PPA encoding model as **simPPA** and the LOC encoding model as **simLOC** in the following.

## 3 Semantic-preference Mapping

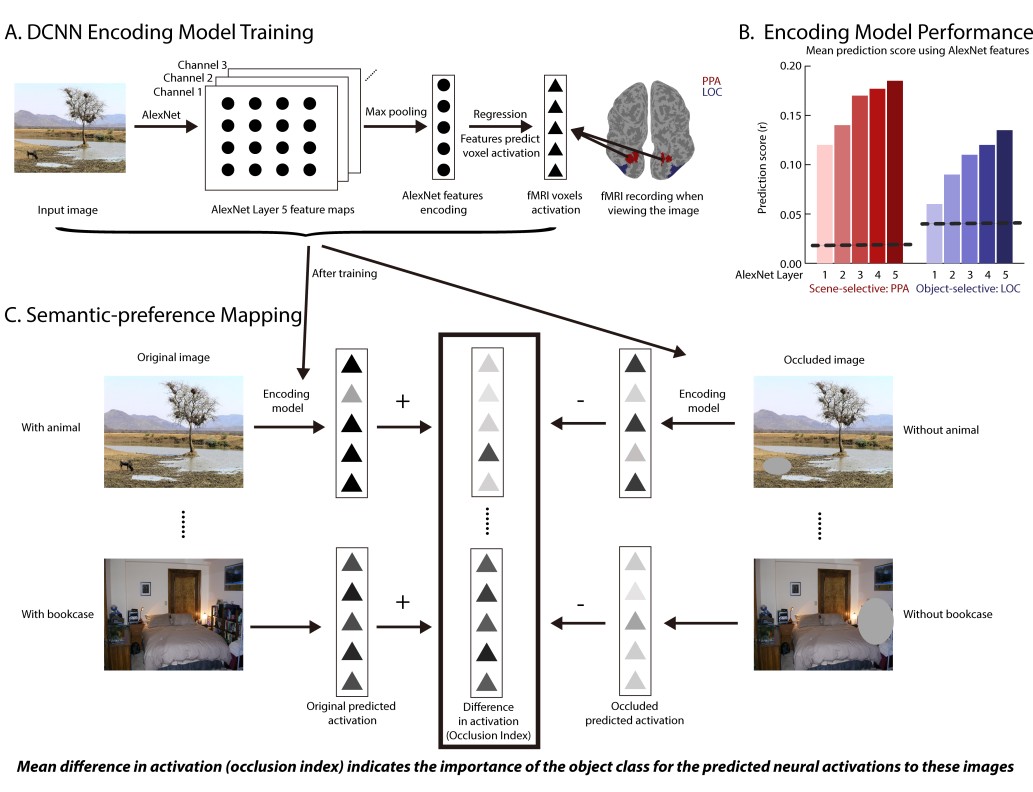

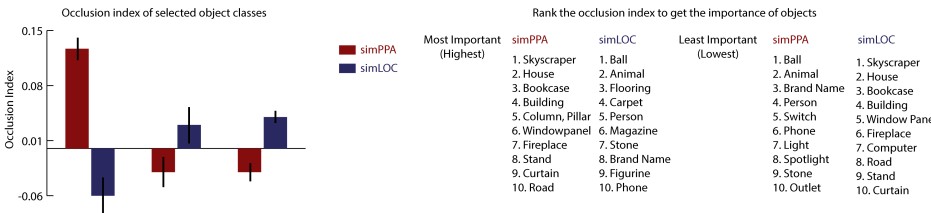

Figure 1: A: Image-computable encoding models were trained on the BOLD5000 dataset using LASSO regression to predict fMRI responses from layer 5 of AlexNet after a global max-pooling procedure. B: In both scene-selective PPA and object-selective LOC, fMRI activations were best predicted from AlexNet layer 5, compared to earlier layers. Dotted lines indicate the noise-ceiling estimate. C: Demonstration of semantic-preference mapping. An image with an occluded object was provided to the encoding model to get the predicted activation, which was compared with the predicted activation of the original image. The occlusion index was the mean difference in activation between the original and occluded images for all instances of an object class. D: Results of semantic-preference mapping. In the left panel, the occlusion index for selected objects in simPPA and simLOC is plotted. Error bars indicate +/-1 s.e.m. across subjects. In the right panel, ranking of the occlusion indices in simPPA and simLOC showed that these ROIs have distinct tuning profiles.

As illustrated in Fig.1C, semantic-preference mapping examines how the predicted activations of DCNN encoding models are affected by the object classes present in an image. The general approach is to systematically occlude a specific object class from each image in a large set of annotated images and calculate the degree to which each object class affects the predicted activations. If a DCNN encoding model is selective for a specific object class, then we would expect to see its predicted activations decrease when that object class is occluded in any image. This procedure was performed using the ADE20K dataset, which has been densely annotated with object segmentation maps [27]. The images in this dataset do not overlap with those in BOLD5000. We examined classes of objects that were present in at least 500 images in ADE20K (85 object classes) to ensure that we had a stable estimate of the semantic-occlusion effects across many instances of each object class. For a target object class and each image containing an object from that class, we first obtained the predicted activation of an encoding model to the original image. We then applied an oval-shaped occluder of random pixel values over the object, fed the occluded image to the encoding model. Occlusion index was obtained by subtracting the predicted activation of the occluded image from the original image and calculated the mean of these differences: $OI = \frac{\sum_{vox} act_{original,vox_i} - act_{occluded,vox_i}}{N_{vox}}$. This way of obtaining occlusion index was similar to obtaining activation contrast in actual fMRI dataset. The oval occluder was the minimum possible size that fully occluded the object (i.e., the occluder covered the entire object segmentation mask), and the edges of the occluder were smoothed to avoid adding high-frequency noise to the image. After repeating this procedure for all images containing the target object class, we calculated the mean occlusion index across images, which captures the degree to which the responses of an encoding model are sensitive to the target object class. We repeated this procedure for all object classes.

simPPA and simLOC showed opposed tuning properties. From Fig.1D left panel (see E for full results), the occlusion indices of selected object classes are shown. simPPA showed a high occlusion index for bookcase and a slightly negative index for animal and ball, while simLOC showed the opposite pattern. In the right panel of Fig.1D, we ranked the occlusion indices of all 85 classes of objects, and found that the object rankings for simPPA and simLOC were in approximately reverse order relative to one another. We also observed that many of the top-ranked object classes for simLOC tended to have curved contours (e.g., ball, animal, person, even flooring and rug has a lot of curvy pattern inside it), whereas the top-ranked object classes for simPPA tended to be more rectilinear (e.g., skyscraper, house, bookcase). This suggests the possibility that one of the factors underlying these object-class preferences may be selectivity for curved versus rectilinear contours. An alternative possibility is that these results simply reflect the occluder size (e.g., simPPA is selective for objects that are larger and thus require larger occluders). However, any potential effects of occluder size are likely minimized by our use of global max-pooling, which discards spatial information from each feature channel. As we show in our follow-up analyses, the occluder size did not account for much variance in these object-class preferences, and our results remained largely the same even after partialing out occluder size (see Appendix D). There are of course many other factors besides curvature and occluder size that might differ between the top-ranked objects for simLOC and simPPA (e.g., real-world size, animacy, spatial stability), and we are systematically exploring these other factors in follow-up analyses. However, our preliminary findings thus far point to curvature as a key explanatory factor. Currently there is a lack of quantitative models for quantifying the presence of curved contours in images. This motivated us to develop an image-computable model of the curvature statistics in natural images, which is the focus of our next set of analyses.

## 4 Interpretation: Curvature

To assess the possibility that curvature is an important dimension underlying the selectivity of our DCNN encoding models, we developed a computational model to detect curved contours in natural images. The objective of this curvature model is to obtain a single curvature index for any given image. The procedure for the curvature model is described in Fig.2A and Appendix B.

We used this curvature model to characterize the contribution of curved contours to the findings of our semantic-preference mapping analysis. First, the image patches that had been occluded during semantic-preference mapping (see Fig.2B) were fed into the curvature model to obtain curvature indices. We then calculated both Pearson and Spearman's correlation between the the curvature indices and the occlusion indices for all object classes. The results in Fig.2C show that there is a strong correlation between curvature indices and occlusion indices for both the simPPA and simLOC. This

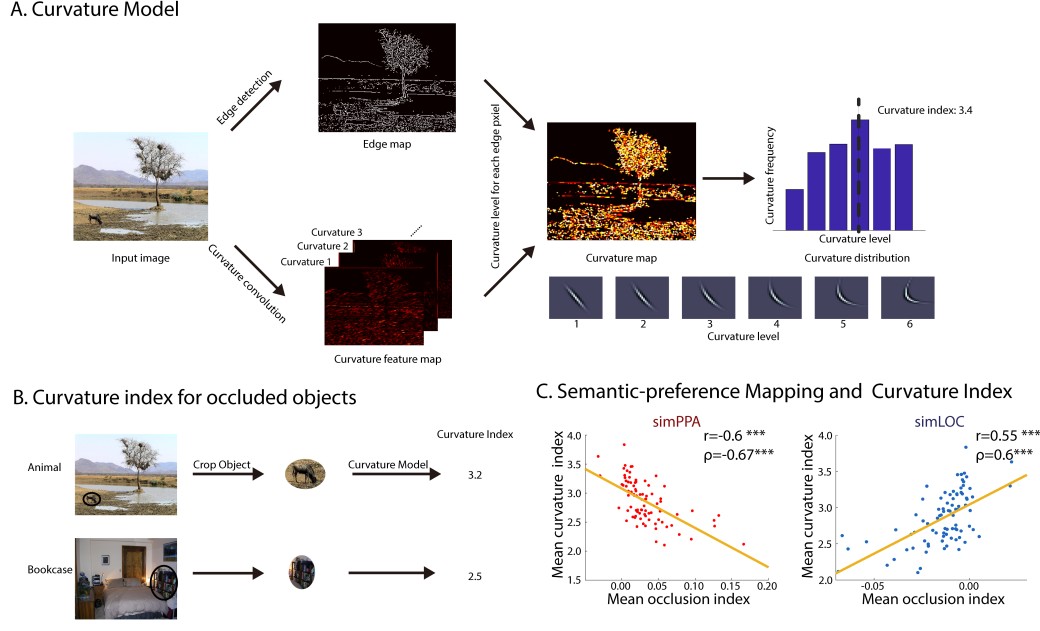

*Semantic-preference mapping reveals curvature as one of the mid-level features encoded in category selective areas*

Figure 2: A: We used a hand-crafted model to compute a curvature index that reflects the presence of curved contours in images. Each image is convolved with a bank of curvature filters that are parameterized according to curvature level and orientation. For each edge pixel, we identify the curvature level that best matches the local image patch. We then obtain a global curvature index by calculating the average curvature level across all edge pixels. B: We computed curvature indices for each object class in our semantic-occlusion procedure. The occluded part of each image was cropped and then fed into the curvature model. For each object class, we calculated the mean curvature index across all images. C: The semantic-occlusion indices and curvature indices were negatively correlated in simPPA and positively correlated in simLOC, suggesting that the semantic-occlusion results may reflect mid-level preferences for rectilinear versus curved contours in simPPA and simLOC.

correlation is positive for simLOC, suggesting a preference for objects containing curved contours. In contrast, the correlation is negative for simPPA, suggesting a preference for objects containing rectilinear contours. To account for the possibility that these effects are confounded with occluder size, we computed partial correlations of the curvature and occlusion indices after partialling out a regressor for occluder size. The results of this partial-correlation analysis were highly similar to our original results, suggesting that occluder size is not a major explanatory factor (see Appendix D). In addition, to rule out the possibility that the shape of occluder was causing the effect, we ran the exact same semantic-preference mapping procedure using rectangular occluder instead of oval occluder, and the occlusion index from the oval occluder and the rectangular occluder were highly correlated (simPPA: $r = 0.87, p < 0.001$, simLOC: $r = 0.76, p < 0.001$). Overall, we observed a striking correlation between the data-driven semantic-selectivity profiles of our ROI models and a simple summary statistic of image curvature. This suggests that when using DCNNs as fMRI encoding models in a large-scale naturalistic stimulus set, curvature emerges as a key property of the mid-level representations that are shared between DCNN feature channels and category-selective regions of ventral visual cortex.

## 5 Discussion

Understanding mid-level visual representations is a longstanding challenge in both neuroscience and computer vision [24, 20]. Here we introduce an approach that combines DCNN-based fMRI encoding models of human visual cortex with large-scale annotated image sets to systematically explore the mid-level representations of human vision. Our semantic-preference mapping procedure leverages dense image annotations to characterize the selectivity of DCNN channels to semantic object classes.

This procedure combines established techniques for receptive-field mapping in DCNNs with an inductive bias for understanding complex mid-level features in the context of the object classes for which they are the most informative. While this procedure was shown to be a powerful tool to explore mid-level feature tuning in DCNN encoding models, the explained variance of our DCNN encoding models is ultimately limited by the noise ceiling of the fMRI data and could be improved with an fMRI dataset that included more stimulus repetitions. Furthermore, a demonstration that our fMRI encoding models can generalize to novel fMRI datasets would provide stronger support for our findings. Future work can extend the current results by relating semantic occlusion findings to other object properties, such as real-world size [14], animacy[11], reachability [9] and memorability [1]. Semantic-preference mapping could also be modified to examine other image features, such as color or spatial frequency, and to explore how these parameters are related to neural representations. We applied this procedure to examine the selectivity profiles of object- and scene-selective ROIs in the ventral stream. Our findings revealed that DCNN encoding models of object- and scene-selective cortex capture a key principle of mid-level representations: selectivity for features with curved versus rectilinear contours. Previous work has demonstrated that curvature is an important feature of representations in primate V4 [5, 8, 21] and that there are patches of curvature-preferring regions along the ventral stream of human and non-human primates [26, 25]. Previous work has also demonstrated a preference for rectilinear junctions in scene-selective regions [18], but see [4]. Here we use a data-driven approach to reveal curvature as an emergent property of mid-level feature tuning in category-selective visual cortex.

## Broader Impact

Semantic-preference mapping is a novel approach for using DCNN encoding models to study neural representation. This approach could be adapted for exploring hypotheses in other cognitive domains, such as audition and language, and it could be useful for exploring representational hypotheses using existing, condition-rich fMRI data sets. In addition, our simple image-computable curvature model could be a tool for researchers to quantify image curvature statistics, without relying on subjective ratings.

## Acknowledgments and Disclosure of Funding

We acknowledge the Maryland Advanced Research Computing Center (MARCC) for providing computational resources that have contributed to the research results reported within this paper. In addition, we thank Alon Hafri, Caterina Magri, Celia Litovsky, Michael McCloskey and Sherry Chien for useful discussion and comments on the manuscript.

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

## Appendix A    LASSO regression

Previous studies [19, 12] suggest that regularization can be helpful for fMRI encoding models to reduce overfitting and to handle correlated regressors. While most studies employ ridge regression (L2 regularization), here we explored how L1 regularization affects encoding model performance. We were interested in L1 regularization as a potential means of learning sparse encoding models that emphasize the DCNN features that are most important for each voxel and ROI. We first evaluated the performance of different regression methods by running encoding-model analyses on the BOLD5000 dataset with 10-fold cross-validation using ordinary least square (OLS) regression (without regularization), LASSO regression (L1 regularization) and ridge regression (L2 regularization). For LASSO and ridge regression, a separate 10-fold cross-validation was performed before assessing performance to determine the best penalty parameters. Because the penalty parameters for LASSO and ridge are learned on the same data that we use for quantifying model performance (using a different cross-validation design), the performance estimates for the regularized models may be slightly biased upwards. However, this is not problematic for our analyses for two reasons. The first reason is that the encoding models perform well above chance even when using OLS regression without regularization, which means that regularization is not required to achieve statistically significant performance. The second reason is that our results and conclusions do not depend on the specific values of the performance estimates. It is already well-established that DCNNs are state-of-the-art models of fMRI responses in visual cortex. The goal of our analyses is to characterize the mid-level representations in DCNN encoding models after they have been fit to fMRI data.

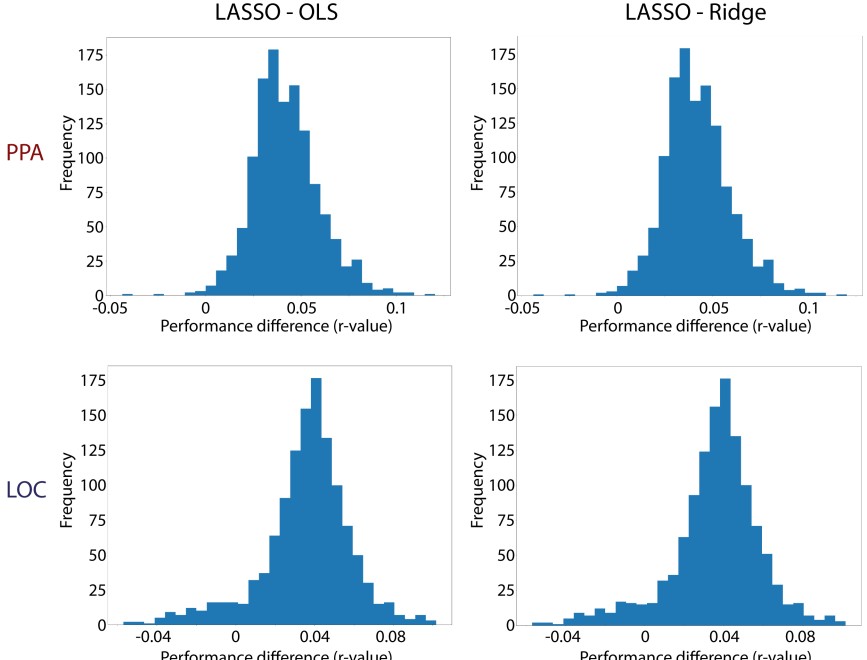

Figure 3: Distribution of performance differences between regression methods. LASSO outperforms both ridge and OLS regression in voxelwise encoding model performance in both PPA and LOC.

We calculated the difference in voxelwise prediction accuracies between pairs of regression methods to create distributions of performance differences (see Fig.3). The results show that LASSO regression outperforms both OLS and ridge regression. One possible explanation for this result is that if only a subset of DCNN features are informative for each voxel, LASSO regression would be most appropriate since it can set the regression weights on uninformative features to zero (thus performing both feature selection and regression).

## Appendix B    Curvature model

The goal of the curvature model is to provide a summary statistic of the curvature of contours in local image regions. Specifically, we calculated a curvature index based on the mean curvature level of edge pixels (see Fig. 2A). We used a filter bank of convolutional kernels for curved contours (see Appendix D for details). The curvature model procedures are as follows:

1. Compute curvature feature maps: The curvature model starts by convolving the curvature filter bank with a grayscale input image. By convolving the image with a curvature filter, we obtained a measure of the similarity between each local image patch and the curvature filter.

2. Edge detection: The grayscale input image is fed into an edge detection algorithm (Roberts) to separate edge pixels from other pixels.

3. Pixel-wise curvature levels: The goal of this step is to compute the curvature level of each edge pixel. A high value for a pixel in a curvature feature map suggests a higher similarity between the corresponding curvature wavelet and the local image patch. Thus, we assigned a curvature level to each edge pixel by finding the corresponding curvature level of the wavelet with the maximum output.

4. Curvature index: We compute the mean curvature level across all edge pixels in the image, and the resulting value is the curvature index for the image.

## Appendix C    Curvature filter

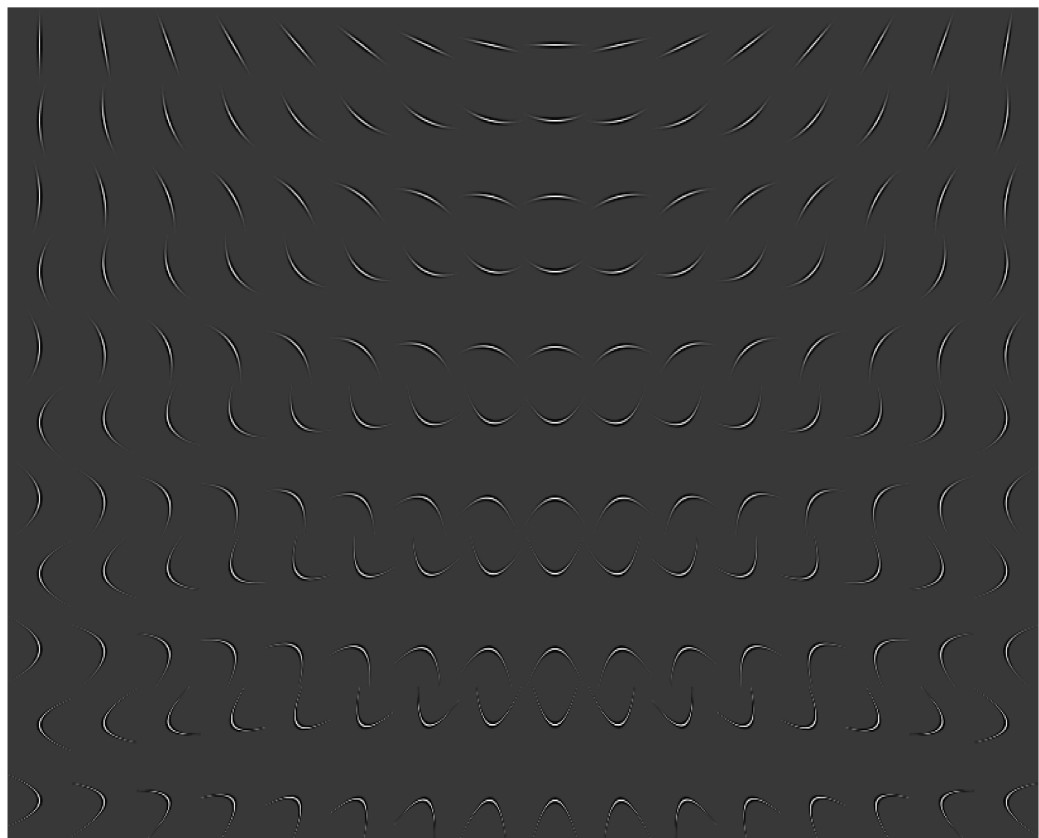

Figure 4: Curvature Filter Bank

In the curvature model, we performed convolution with curvature-detector filters and obtained curvature indices for each pixel [16]. We created a curvature filter bank Fig.4. Each curvature filter can be thought as a curved-contour detector with a particular orientation and curvature level. We convolved each image with 176 different curvature filters that spanned 16 orientation levels and 11 curvature levels (5 concave levels, 5 convex levels and 1 straight level) and identified the most strongly activated curvature level across all orientations at each pixel. A curvature filter is built by combining a rotated and curved complex wave function ($F$) and a rotated and curved Gaussian function ($G$). This curvature filter idea is borrowed from the construction of the Gabor wavelet filter, which consists of a sinusoid function and a gaussian function. A single curvature filter is parameterized by six variables, including frequency ($f$), orientation ($\theta$), curvature ($c$), size ($s$), and scale of the gaussian filter in x ($\sigma_x$) and y ($\sigma_y$) direction. Each filter can be composed by the following mathematical formulas:

$$B(x,y) = G(x,y) \cdot (F(x,y) - bias) \tag{1}$$

$$G(x, y) = exp\left[-\frac{f^2}{2} \cdot \left(\frac{(x_c + c \cdot s_s^2)^2}{\sigma_x^2} + \frac{x_s^2}{s^2 \cdot \sigma_y^2}\right)\right] \tag{2}$$

$$F(x, y) = exp(i \cdot f \cdot (x_c + c \cdot x_s^2)) \tag{3}$$

$$x_c = x \cdot cos\theta + y \cdot sin\theta \tag{4}$$

$$x_s = -x \cdot sin\theta + y \cdot cos\theta \tag{5}$$

$$bias = e^{-\frac{\sigma_x}{2}} \tag{6}$$

## Appendix D  Occluder size

We tested whether occluder size could explain the results of the semantic-preference mapping procedure. In this analysis, we quantified the occluder size by counting the number of occluded pixels. Table 1 shows the partial correlation of the mean occlusion index with both mean occluder size and mean curvature index for each object class.

Table 1: Partial correlation of semantic-preference mapping results with occluder size and curvature. *** indicates $p < 0.001$

|  | Partial Correlation | |
| --- | --- | --- |
| ROI | Occluder size | Curvature index |
| PPA | 0.16 | -0.55 *** |
| LOC | -0.14 | 0.53 *** |

In both PPA and LOC, occluder size did not significantly correlate with the semantic-preference mapping results when curvature was taken into account. On the contrary, curvature indices were significantly correlated with the semantic-preference mapping result in both PPA and LOC after taking occluder size into account. In summary, this suggests that occluder size is not a strong explanatory factor for the semantic-preference mapping results.

## Appendix E  Full semantic-preference mapping result

Table 2: Semantic-preference mapping results of PPA and LOC

|  | PPA | LOC |
| --- | --- | --- |
| Most important | Skyscraper | Ball |
|  | House | Animal |
|  | Bookcase | Flooring |
|  | Building | Rug |
|  | Computer | Person |
|  | Windowpanel | Magazine |
|  | Fireplace | Rock, Stone |
|  | Stand | Brand Name |
|  | Curtain | Figurine |
|  | Road, Route | Telephone |
|  | Swivel Chair | Switch |
|  | Blind, Screen | Light |
|  | Dresser | Ashcan |
|  | Palm, Palm Tree | Fluorescent |
|  | Sink | Outlet |
|  | Desk | Shoe |
|  | Stove | Spotlight, Spot |
|  | Sky | Pot, Flowerpot |
|  | Column | Bicycle, Bike, Wheel, Cycle |
|  | Chandelier | Plaything, Toy |

Railing, Rail
Painting, Picture
Armchair
Sea
Poster
Coffee Table
Food, Solid Food
Earth, Ground
Stool
Sidewalk, Pavement
Table
Television
Towel
Plant, Flora, Plant Life
Fence, Fencing
Field
Bannister
Truck, Motortruck
Signboard, Sign
Sofa, Couch, Lounge
Stairway, Staircase
Awning
Pillow
Grass
Air Conditioner
Boat
Path
Can, Tin, Tin Can
Traffic Light
Bucket, Pail
Ashcan
Car
Floor, Flooring
Book
Mountain, Mount
Flag
Monitor, Monitoring Device
Basket, Handbasket
Shrub, Bush
Stairs, Steps
Plaything, Toy
Van
Jar
Umbrella
Candlestick, Candle Holder
Shoe
Minibike, Motorbike
Streetlight, Street Lamp
Magazine
Bicycle, Bike, Wheel, Cycle
Rug
Pot, Flowerpot
Fluorescent
Figurine, Statuette
Glass, Drinking Glass
Outlet
Stone
Spotlight
Light
Telephone
Switch
Person
Brand Name
Animal

Glass, Drinking Glass
Boat
Minibike, Motorbike
Stairs, Steps
Path
Van
Jar
Pillow
Candlestick, Candle Holder
Bucket, Pail
Table
Awning
Umbrella
Basket, Handbasket
Sofa, Couch, Lounge
Car
Towel
Shrub, Bush
Air Conditioner
Streetlight, Street Lamp
Flag
Mountain, Mount
Can, Tin, Tin Can
Monitor, Monitoring Device
Book
Traffic Light
Bannister
Fence, Fencing
Plant, Flora, Plant Life
Stairway, Staircase
Television
Coffee Table
Earth, Ground
Sidewalk, Pavement
Armchair
Poster
Stool
Food, Solid Food
Signboard, Sign
Truck, Motortruck
Sink
Grass
Desk
Painting, Picture
Dresser
Field
Chandelier, Pendant, Pendent
Railing, Rail
Stove
Sky
Swivel Chair
Column, Pillar
Blind, Screen
Palm, Palm Tree
Sea
Curtain
Stand
Road, Route
Computer
Fireplace
Windowpanel
Building
Bookcase
House

| Least important | Ball | Skyscraper |
| --- | --- | --- |

