# OpenReview forum: "Curvature as an Organizing Principle of Mid-level Visual Representation: A Semantic-preference Mapping Approach"
_NeurIPS.cc/2020/Workshop/SVRHM — SVRHM@NeurIPS Poster_

### Official Review · AnonReviewer2 · 2020-10-27
**Too many layers of abstraction with low fidelity to trust the results**

**Rating:** 3
**Confidence:** 5

**Review:**

I'm struggling a bit with the categorization of two deep category specific areas like PPA and LOC being described as mid-level areas for which the selectivity properties of their neurons is not tied closely to interpretable concepts.  Their names (Parahippocampal Place Area and Object-selective Lateral Occipital Cortex) and location in the visual hierarchy suggest quite the opposite (whereas areas v2 and v4 are more aptly described as mid level areas that defy easy concept categorization).

I'm also struggling with the logic behind the claims in this paper.  A Deep Conv net is fit to Voxel-wise fMRI responses, and one layer of the network predicts brain responses with a Pearson-r of < .2.  Then a 'context ablation' study is carried out on the network alone and the "occlusion indexes" derived from this study are attributed to LOC and PPA without any measurement of those areas?  In addition, broad classes of shapes are derived supposedly corresponding to the shapes in the occlusion study (though this doesn't appear to be totally clear either, one list starts with Ball which is used to indicate preference for a preference for curves, but flooring and carpet and magazine are also included in the top selectivity, and it is not clear those are curved?), let alone that the oval mask applied in the ablation study fits the broad "curvature" category which seems like a large confound.

Then following on this, a handcrafted curvature model is applied to the images (and specifically quantified on the areas of the image that are masked in the ablation study) and the curvature index of that patch is correlated with the occlusion index of the DCNN (and the result attributed again to PPA and LOC though this is never actually measured in the brain).  Which I am again struggling to understand why we should believe this is indicative of encoding within the actual brain areas or even that it may be the most salient of the selective characteristics.  Why not use the curvature model on the original dataset to see if it predicts differences in firing rates in the actual fMRI data?

There is definitely benefit in utilizing DCNNs in conjunction with fMRI data to build image computable models of the visual system and to help elucidate underlying representation sensitivity, but it seems the authors didn't leverage the power of these tools correctly.  The methodology presented within treats the DCNN like a black box, similarly to how they would treat the brain, and looks for loose measures of aggregated response changes.  These methods could have all been presented to actual humans in fMRI to attempt to pull out more convincing illustrations of the underlying claims.  As it stands, they were applied to a network that sort of predicts responses in those areas, which is less than compelling.

---

### Official Review · AnonReviewer3 · 2020-10-28
**A well-presented and clever approach to generating hypotheses about brain regions**

**Rating:** 9
**Confidence:** 4

**Review:**

The paper cleverly leverages the interrogability of DNNs in order to better understand brain regions. While only a relatively small number of images can be shown to humans during fMRI scanning, a vast number can be shown to a DNN. The current work exploits this by fitting a DNN-feature-based encoding model from human fMRI data, and then exploring the preferences of the encoding model to a large number of systematically manipulated images.

The authors train an encoding model to predict voxel activations in PPA and LOC (separately) from the (max-pooled) activations of feature maps in Alexnet layer 5 (meaning there are 192 features in the encoder, I believe). Then these encoding models are used to predict PPA/LOC activations for a new set of images, belonging to 85 different categories. For every image, brain activations are predicted once for the full version of the image, and once for a version in which the key object has been occluded with pixel noise. The difference in predicted activations is calculated and gives an "occlusion index" of how much the encoding model predicts that the brain region in question cares about the presence or absence of this object. The model's rankings of different object types has good face validity - it predicts PPA will care most about the presence of Skyscrapers, Houses, Buildings etc, while LOC will care most about Balls, Animals, Persons etc. Based on some intuitive guesswork about what might differentiate these groups of things, the authors then hypothesise that PPA is responding strongly to objects with rectilinear edges, and LOC to objects with curvy edges. They quantify this by implementing a local curvature model, and showing that there is indeed a reasonably strong correlation between local curvature within objects and occlusion index (how much LOC/PPA are predicted by the encoding model to care about the presence or absence of that object). Strictly speaking, this final analysis only tests how well curvature can account for the responses of the *encoding model*, not of LOC/PPA themselves, since no new fMRI data are collected. An additional, independent, fMRI dataset would be needed to test how well this hypothesis predicted brain data. However, I think the approach is a useful and interesting one to explore data and generate testable hypotheses.

Pros:
+ Excellent detailed informative figures making complex multi-step methods clear.
+ Independent set of images used for the semantic preference mapping from those used in the fMRI experiment and encoding model fitting.
+ Thorough exploration of a large set of object classes (eighty-five classes), which bolsters confidence that curvature is not an accidental confound within the tested classes, but something fairly general across visual classes.
+ Potentially widely useful data-driven approach to explore factors influencing neural responses.

Cons:
- It's not fully clear how prediction score and noise ceiling for fMRI encoding models were computed (e.g. was cross-validation done over images or over runs, with the possibility of repeated images? How was the noise ceiling defined?). All encoding models perform substantially above the "noise ceiling", which implies that the noise ceiling is not defined in such a way that it actually indicates the strict "ceiling" on explainable variance (as it does under the definition used in, for example, representational similarity analysis)
- Not specified how the Occlusion Index is calculated - "difference in activation" probably implies Euclidean distance between activation vectors? But could also imply a univariate measure like difference in mean activation level between vectors.
- Would of course have been ideal to gather new fMRI data at the final stage, in order to provide an *experimental* test of the curvature hypothesis. But there is already a substantial amount of work in the current submission.

---

### Official Review · AnonReviewer1 · 2020-10-29
**Semantic-preference mapping review**

**Rating:** 8
**Confidence:** 3

**Review:**

Voxelwise encoding models allow researchers to predict fMRI responses using a linear combination of features from CNNs. Here, the authors develop a new method – semantic-preference mapping – to understand how predicted fMRI activations differ when specific objects are removed from the image input. They find that object-removals that change the predicted activations of PPA are different from object-removals that change the predicted activations of LOC. The authors observed that curvature is one potential feature explaining these differences – removing rectilinear objects affected predicted activations of PPA while removing curved objects affected predicted activations of LOC. To test this hypothesis, the authors developed a computational model which allowed for the estimation of curvature for any given image. This measurement of curvature verified their intuition – the greater the rectilinearity of the occluded object the more it affected predicted activations of PPA and the less it affected predicted activations of LOC.

The authors show how semantic-preference mapping provides researchers with an interesting new tool for seeing how the predictions of encoding models are dependent on different image properties. One could investigate whether your encoding model predicts changes in activations within a brain region when removing objects with any number of measurable features. The contribution of a computational model for measuring curvature is another positive aspect of the paper.

My main concern with this method surrounds how results are interpreted. In Section 3, the authors make direct arguments about the tuning properties of PPA and LOC when they are actually comparing predicted activations to new images made by their encoding models within the two regions. Without actual fMRI responses to the ADE20K dataset, we cannot verify whether the occlusion indices accurately predict fMRI responses. It is possible that the encoding model fit on the BOLD5000 dataset does not generalize to the ADE20K dataset or to the dataset with added occlusions. Collecting fMRI responses to the new images would allow for predictions made from semantic-preference mapping to be verified. This is necessary to make direct arguments about tuning properties of PPA and LOC rather than just arguments about the predictions of your encoding model.

In the discussion, the authors claim: “our study is the first to use a data-driven approach to reveal curvature as a key representational dimension of category-selective visual cortex”. This claim feels unearned given that Long, Yu, and Konkle (2018) have shown that curvature ratings predict the representational geometry of OTC in response to both objects and texforms.

Less important suggestions:
The accuracy of encoding models is dependent on the level of reliability of the fMRI data. As you show in Fig1, the noise ceiling of the BOLD5000 dataset is quite low, hampering the ability to create encoding models that are highly predictive of fMRI responses. I am surprised you were able to fit encoding models that made predictions so much higher than the noise ceiling. If the noise ceiling reflects the level of signal in the data, I am not sure how your models were able to exhibit so much higher correlations. My understanding is that the event-related design of the BOLD5000 dataset hampered the reliability of the data. If possible, I would suggest trying this method using a dataset with higher reliability, though I also acknowledge that very few datasets exist like this yet, so I don’t hold the low reliability of the BOLD5000 dataset against the authors. Kendrick Kay’s upcoming Natural Scenes Dataset may be one such dataset, as he has reported high reliability in his Algonauts talk.

The authors show the objects with the highest and lowest occlusion indices for predicted-PPA and predicted-LOC. They note that the object rankings are in approximately reverse order for the two regions. Reporting the Spearman correlation between the two object rankings would be helpful for more fully backing up this claim.

---

### Public Comment · ~Shi_Pui_Donald_Li1 · 2020-12-07
**Response to reviews**

We thank the reviewers for their helpful comments. We have made several revisions based on these comments.

All reviewers brought up the possibility that encoding models trained on the BOLD5000 data may not be able to generalize to the ADE20K stimuli used for our in silico experiments. This is a valid point, and we have added a comment in the Discussion about the utility of testing generalization to a completely new fMRI dataset. Nonetheless, we would also like to point out that high-throughput, in silico experiments, like those reported here, are important tools for computational neuroscientists, especially when investigating deep neural networks as representational models of the brain. Neuroscientists continue to debate the theoretical relevance of deep neural networks, due largely to the black-box nature of their internal representations. Gaining deeper theoretical insights into these models will require large-scale, in silico experiments. Running the same experiments in human subjects at this scale is not feasible. Furthermore, this approach allows us to develop rigorous, quantitative investigations of the relationship between mid-level feature tuning and the higher-level properties of objects and scenes.

To make it clear that our in silico experiments are investigating the nature of the encoding models fit to the PPA and LOC rather than additional data for these regions, we now use the terms simPPA and simLOC when describing the findings for these encoding models.

Reviewers 1 and 2 had a question about how the performance of the encoding model can exceed the noise ceiling. Our “noise ceiling” estimate is the split-half reliability of fMRI responses for a subset of the data and is thus not a hard ceiling. We have clarified this point in the model training section.

Related to the semantic-preference mapping procedure, Reviewer 2 pointed out that the way we calculated the occlusion index was not clear. We have clarified this procedure by including the equation in the main text.

Reviewer 1 suggested reporting Spearman correlation of the occlusion index and the curvature index in addition to Pearson correlation. We have now added the Spearman correlation in Figure 2, which shows the same effect as the Pearson correlation. We also included the full occlusion index ranking in the Appendix.

Reviewer 3 raised a concern that the oval shape of the occluder could have influenced the results. We found that the results are not contingent on the particular shape of the occluder. For example, rectangular occluders produce highly similar results. We now report this in the Curvature Interpretation.

---

### Public Comment · ~Shi_Pui_Donald_Li1 · 2021-01-14
**Curvature model code available**

The code of the curvature model is now available at : https://github.com/shipui2005/Curvature-Model.git

---

### Decision · Program_Chairs · 2020-11-02

Accept (Poster)